

# Six-year mortality in a street-recruited cohort of homeless youth in San Francisco, California

Colette L. Auerswald[1], Jessica S. Lin[2] and Andrea Parriott[3]

[1] University of California Berkeley-University of California at San Francisco Joint Medical Program, School of Public Health, University of California Berkeley, Berkeley, CA, United States
[2] School of Public Health, University of California Berkeley, Berkeley, CA, United States
[3] Phillip R. Lee Institute for Health Policy Studies, University of California at San Francisco, San Francisco, CA, United States

## ABSTRACT

**Objectives.** The mortality rate of a street-recruited homeless youth cohort in the United States has not yet been reported. We examined the six-year mortality rate for a cohort of street youth recruited from San Francisco street venues in 2004.

**Methods.** Using data collected from a longitudinal, venue-based sample of street youth 15–24 years of age, we calculated age, race, and gender-adjusted mortality rates.

**Results.** Of a sample of 218 participants, 11 died from enrollment in 2004 to December 31, 2010. The majority of deaths were due to suicide and/or substance abuse. The death rate was 9.6 deaths per hundred thousand person-years. The age, race and gender-adjusted standardized mortality ratio was 10.6 (95% CI [5.3–18.9]). Gender specific SMRs were 16.1 (95% CI [3.3–47.1]) for females and 9.4 (95% CI [4.0–18.4]) for males.

**Conclusions.** Street-recruited homeless youth in San Francisco experience a mortality rate in excess of ten times that of the state's general youth population. Services and programs, particularly housing, mental health and substance abuse interventions, are urgently needed to prevent premature mortality in this vulnerable population.

## INTRODUCTION

Homeless youth experience a disproportionate risk for morbidity, including, but not limited to, HIV and other sexually transmitted infections, Hepatitis B and C, and psychiatric disorders (*DeMatteo et al., 1999*; *Ensign & Gittelsohn, 1998*; *Hahn et al., 2001*; *Kennedy, 1991*; *Larkin Steet Youth Services, 2014*; *Noell et al., 2001*; *Roy et al., 1999*). Homeless adults have been shown to experience increased mortality (*Hwang, 2000*; *Nordentoft & Wandall-Holm, 2003*; *Alstrom, Lindelius & Salum, 1975*; *Babidge, Buhrich & Butler, 2001*; *Barrow et al., 1999*; *Beijer et al., 2011*; *Cheung & Hwang, 2004*; *Hibbs et al., 1994*; *Hwang et al., 1997*; *Metraux et al., 2011*; *Nielsen et al., 2011*; *Shaw & Dorling, 1998*; *Baggett et al., 2013*; *Nusselder et al., 2013*). This elevated risk of mortality may be especially high amongst homeless youth. Table 1 summarizes the existing published findings regarding street youth mortality rates and standardized mortality rates in Europe, North America, and Australia. Of the studies that have examined rates of mortality among adult homeless populations,

Corresponding author
Colette L. Auerswald,
coco.auerswald@berkeley.edu

eleven have presented data for a youth subgroup within the sample (*Hwang, 2000*; *Nordentoft & Wandall-Holm, 2003*; *Alstrom, Lindelius & Salum, 1975*; *Babidge, Buhrich & Butler, 2001*; *Barrow et al., 1999*; *Hibbs et al., 1994*; *Hwang et al., 1997*; *Metraux et al., 2011*; *Nielsen et al., 2011*; *Shaw & Dorling, 1998*; *Nusselder et al., 2013*). Of those reporting age- and gender-specific standardized mortality ratios (SMR), youth SMRs ranged from 2.1 to 37.32, generally higher than the SMRs found amongst older participants in the same studies (*Hwang, 2000*; *Nordentoft & Wandall-Holm, 2003*; *Alstrom, Lindelius & Salum, 1975*; *Babidge, Buhrich & Butler, 2001*; *Barrow et al., 1999*; *Hibbs et al., 1994*; *Hwang et al., 1997*; *Roy et al., 2010*; *Roy et al., 2004*).

Of note, a census-based study of life expectancies in Canada found that a 25-year old male living in shelters, rooming houses, or hotels had a 32% chance of surviving to the age of 75, as compared to 51% of housed males living in the lowest fifth income bracket (*Hwang et al., 2009*). A national registry-based study of homeless persons in Denmark found that the remaining life expectancy for individuals who first accessed shelter between the ages of 15–24 was 38.7 years for men and 47.4 years for women, compared to 60.3 and 64.8 years respectively for the general population of Danish men and women (*Nielsen et al., 2011*). Investigators in Montreal have focused specifically on street youth, conducting two five-year prospective cohort studies with youth initially recruited from service agencies at ages 14–25 (*Roy et al., 2010*; *Roy et al., 2004*; *Roy et al., 1998*). Roy and colleagues found an SMR of 15.3 in their first cohort tracked from 1995–2000, and a subsequent 79% mortality rate decrease to an SMR of 3.0 in their second cohort tracked from 2001–2006, attributed to an improvement in local services to help the homeless and injection drug using populations.

Although the United States and Canada are geographically and economically similar, the national health care system and broader social safety net in Canada offers a substantially different context and resources for youth homelessness in the two countries. Thus, mortality rates found in Canada may not be comparable to those in the US. Of the four existing US-based studies that presented youth-specific mortality data, three were conducted prior to 2000. All four were retrospective studies of homeless adults including youth over the ages of 15 or 18, whose names were collected from shelter or service utilization records (*Barrow et al., 1999*; *Hibbs et al., 1994*; *Hwang et al., 1997*; *Metraux et al., 2011*).

A limitation of all of the studies listed in Table 1 is the use of convenience samples recruited from service providers, such as shelters and clinics. A challenge to recruiting street youth is that they are less likely to access services than homeless adults, making service-based sampling less likely to be representative of a homeless youth population than for adults. Our prior research suggests this may be particularly true of homeless African American youth, who were far less likely to access services for homeless youth than were their white counterparts (*Hickler & Auerswald, 2009*). Similarly, we found in another study that homeless youth recruited during street outreach had higher rates of high-risk sexual and drug-related behaviors than youth recruited from local clinics for homeless youth (*Auerswald et al., 2007*). Given the challenges to recruiting members of a hard-to-reach population that cannot be enumerated using conventional methods, other approaches, including venue-based sampling, have been proposed to recruit a purposive

Auerswald et al. (2016), *PeerJ*, DOI 10.7717/peerj.1909

**Table 1  Mortality rates, person-years of observation, and standardized mortality rates in published studies (or study subsets) of homeless youth.**

| | Age range | Subject recruitment | Follow up (years, person years) | Mortality rate (per 1,000 person-years) | SMR (CI) | Reference population |
|---|---|---|---|---|---|---|
| **United States** | | | | | | |
| Philadelphia (*Hibbs et al., 1994*) | 15–34 | Shelter/service record review | 3 (12,481) | 6.03 | 3.8 (2.8–5.7)[a] | Philadelphia general population |
| Boston (*Hwang et al., 1997*) | 18–24 | Health care services record review | 6 | Males: 5.34 Females: 1.96 | Males: 5.9 (2.1–17.0)[a] Females: 11.8 (4.2–33.1)[a] | Boston general population |
| New York (*Barrow et al., 1999*) | 20–24 | Random recruitment at shelters | 7 | Males: 6.85 Females: no deaths | Males: 4.2 (0.4–11.9)[a] Females: no deaths | US population |
| New York (*Metraux et al., 2011*) | 20–24 | Shelter record review | Mean: 11.4 (975,916) | Males: 4.63 Females: 2.87 | NR | |
| **Canada** | | | | | | |
| Toronto (*Hwang, 2000*) | 18–24 | Shelter record review | 2 (22,958) | 4.21 | 8.3 (4.4–15.6)[b] | Toronto general population |
| Montreal (*Roy et al., 2004*) | 14–25 | Service-based recruitment | 6 (2,822) | 9.21 | 11.4 (7.4–16.7)[c] Males: 11.1 (6.9–16.8)[a] Females: 13.5 (3.6–34.5)[a] | Quebec province general population |
| Montreal (*Roy et al., 2010*) | 14–25 | Service-based recruitment | 5 | 1.9 | 3.0 (1.0–6.9)[c] | Quebec province general population |
| **Europe** | | | | | | |
| Stockholm (*Alstrom, Lindelius & Salum, 1975*) | 20–29 | Social/temperance aid service record review | 3 | NR | 9.0[b] | Stockholm male population |
| London (*Shaw & Dorling, 1998*) | 16–29 | Death records | NR | 40.1 | 37.32 (20.38–62.63)[c] | England and Wales male population |
| Copenhagen (*Nordentoft & Wandall-Holm, 2003*) | 15–24 | Government record review | 10 | NR | Males: 13.3[a] Females: 28.5[a] | Copenhagen general populaiton |
| Rotterdam (*Nusselder et al., 2013*) | 20–29 | Service record review | 10 (17,619) | Males: 9.11 Females: no deaths | NR | Rotterdam general population |
| **Australia** | | | | | | |
| Sydney (*Babidge, Buhrich & Butler, 2001*) | 20–29 | Shelter/psych service record review | 10 (range: 7–11) | NR | Males: 3.51 (1.29–7.64)[a] Females: 16.67 (0.42–92.02)[a] | New South Wales general population |

**Notes.**
[a] Age-adjusted.
[b] Age-adjusted, males only.
[c] Age and sex-adjusted.

sample (*Auerswald et al., 2004*; *Minnis et al., 2002*; *MacKellar et al., 1996*; *Muhib et al., 2001*; *Stueve et al., 2001*; *Kral et al., 2010*).

We undertook the current study to assess the standardized mortality ratios for a six-year prospective cohort of homeless youth recruited entirely from street sites, using venue-based sampling in San Francisco in 2004.

## METHODS

The Street Youth in Social Environments (Street Y-SE) study was a longitudinal study which examined the relationships among street culture, social networks and STI/HIV risk in San Francisco homeless youth. Recruitment, sampling, and study methods have been previously described (*Hickler & Auerswald, 2009*; *Parriott & Auerswald, 2009*). We employed venue-based sampling, a method of accessing both service-engaged and non-engaged hard-to-reach populations, such as homeless youth, by recruiting them at the venues where they spend time (for example, street corners or parks) (*Auerswald et al., 2004*; *Minnis et al., 2002*; *MacKellar et al., 1996*; *Muhib et al., 2001*; *Stueve et al., 2001*; *Kral et al., 2010*). Prior to enrollment, we conducted a mixed qualitative–quantitative assessment of venues to inform the selection of our recruitment sites. Based on street observations and ethnographic interviews with homeless youth, providers and outreach workers, a list of 62 preliminary venues was compiled. Brief street interviews were subsequently conducted at each venue to characterize the volume and composition of the youth population at each site. Based on these findings, the final list of venues was narrowed to 28, based on gender composition, number of youth and redundancy. Our approach to venue selection was adapted from a prior study (*Auerswald et al., 2004*; *Minnis et al., 2002*).

Because youth are often intermittently homeless, eligibility was limited to youth who were 15–24 years of age and who had experienced unstable housing in the prior six months, defined as "having to stay two nights or more in a place that is not your home because you could not stay in your home or you did not have a home, including having to stay in one of the following places: a shelter, outdoors, a squat, with a stranger or someone you did not know well, a car, on public transportation, or SRO/hotel." Exclusion criteria included being actively under the influence of alcohol or illicit substances, or being too distressed or agitated to participate in the interview. Youth provided written informed consent for their participation.

Study participants completed audio computer-assisted self-interviews (ACASI) at the time of enrollment. Interviews were conducted in English. Gender (male, female, or transgender), race/ethnicity, and age were self-reported. Youth were allowed to report up to six racial categories (Black/African American, White/European American, Native American/Alaskan, Latino/Hispanic, Pacific Islander/Polynesian, Asian/Asian–American, Mixed) or "Don't Know" or "Refuse." Interviews also recorded information regarding characteristics such as youth's housing history, service or health care utilization, sexual and substance use patterns, education, and experiences while on the street (*Hickler & Auerswald, 2009*; *Parriott & Auerswald, 2009*).

Deaths were tracked via the National Death Index (NDI), a nationwide index of death certificate data, through the end of 2010. Causes of death were determined by

NDI-Plus. Methods for identifying a match are described in detail on the NDI website (http://www.cdc.gov/nchs/ndi.htm). The index was queried and matches were made using legal names, aliases, dates of birth, and Social Security numbers, as reported by study participants. Follow-up time began on the date of enrollment into the study, and ended either at death or on December 31, 2010.

To calculate a standardized mortality ratio, youth were divided into 32 strata based on age (15–24 or 25–34 years old), sex (male or female, as no street-recruited youth identified as transgender) and race/ethnicity. We employed these age strata as they are those employed in the reporting of California mortality rates (*California Department of Public Health, 2016*). For this analysis, all persons who reported more than one racial category or chose "mixed" as their primary racial identity were classified as "two or more races" and all persons who reported "don't know" or "refuse" or left the race/ethnicity question blank were considered to be of "unknown" race or ethnicity. As there were no Asian/Asian American youth in our sample, only seven race/ethnicity categories were included. For stratum-specific comparison mortality rates, we used the mean mortality rate for California for the years 2004 through 2010 (*California Department of Public Health, 2016*). For the four strata of youth of unknown race/ethnicity, we employed the overall California youth population of the same age and sex as our reference population for the SMR. SAS version 9.3 was used to calculate stratum specific follow-up time, and expected deaths were calculated using Excel. The SMR and Fisher's exact 95% confidence intervals were calculated using OpenEpi version 3.03 (*Dean, Sullivan & Soe, 2014*).

We examined the bivariate relationship of demographic and behavioral variables to mortality using Fischer's 2-sided exact test.

The protocol was approved by the Committee for Human Subjects of the University of California at San Francisco (Study Number 11-05254).

## RESULTS

Our street-recruited sample included 218 youth. One hundred and forty three (65.6%) were male and 75 (34.4%) were female. The mean sample age at baseline was 20.5 years, with a standard deviation of 2.1 years. The sample was 50.5% non-Latino/white, 14.7% non-Latino Black, 26.6% multi-racial, 2.8% Native American, 1.4% Latino, 0.9% Pacific Islander, and 3.2% unknown.

The cohort accrued a total of 1,150.8 years of follow-up. There were a total of 11 deaths for a death rate of 9.6 deaths per thousand person-years of follow-up. Of the 11 deaths, 8 were male and 3 were female; 8 were non-Latino/Hispanic white and 3 were of mixed race. The age, race, and sex standardized mortality ratio for the entire sample was 10.6 (95% CI [5.3–18.9]). The gender specific SMRs were 16.1 (95% CI [3.3–47.1]) for females and 9.4 (95% CI [4.0–18.4] for males.

Six out of eleven of the deaths occurred outside of California. Causes of death by ICD-10 code (the information provided by the NDI database) are listed in Table 2. The majority of deaths were suicides and/or alcohol- or drug-related. Our definition of alcohol- or drug-related deaths corresponds with that of the Center for Disease Control's Division of Vital Statistics (*Kochanek et al., 2011*).

**Table 2  Characteristics of study cohort survivors and decedents.**

| | Survivors (*n* = 207)[*] (*n*; % of responses) | Decedents (*n* = 11)[*] (*n*; % of responses) |
|---|---|---|
| Gender[a] | | |
| Male | 135 (65.2%) | 8 (72.7%) |
| Female | 72 (34.8%) | 3 (27.3%) |
| Race/ethnicity | | |
| Non-Latino/Hispanic White | 102 (49.3%) | 8 (72.7%) |
| Non-Latino/Hispanic Black | 32 (15.5%) | 0 |
| Native American/Alaskan | 6 (2.9%) | 0 |
| Latino/Hispanic | 3 (1.4%) | 0 |
| Pacific Islander/Polynesian | 2 (1.0%) | 0 |
| Asian/Asian American | 0 | 0 |
| 2 or more races | 55 (26.6%) | 3 (27.3%) |
| Unknown | 7 (3.4%) | 0 |
| Age at recruitment | | |
| 15–17 | 19 (9.2%) | 0 |
| 18–20 | 83 (40.1%) | 4 (36.4%) |
| 21 or older | 105 (50.7%) | 7 (63.6%) |
| Age at 1st unstable housing | | |
| 14 or younger | 73 (35.3%) | 6 (54.5%) |
| 15–17 | 82 (39.6%) | 4 (36.4%) |
| 18–20 | 46 (22.2%) | 1 (9.1%) |
| 21 or older | 6 (2.9%) | 0 |
| Any services in last 3 mo. | 122 (58.9%) | 5 (45.5%) |
| Any healthcare in last 3 mo. | 63 (30.4%) | 4 (36.4%) |
| Survival sex | 36 (17.4%) | 2 (18.2%) |
| Ever injected drugs[**] | 75 (36.2%) | 7 (63.6%) |
| MSM[b] | 14 (13.5% of males) | 0 (0% of males) |
| High school graduate (only among respondents > 18 yo) | 109 (52.7%) | 5 (50.0%) |

**Notes.**

[a]None of the youth in the sample identified as transgender.

[*]There were no statistical differences between the characteristics of survivors and decedents at the $p < .05$ or $p < .1$ level.

[**]$p = .11$.

[b]Though there were not missing data for other questions, there were missing data for men reporting having sex with men (39/143). However, all male decedents responded.

Table 3 summarizes the demographics and behavioral characteristics of the decedents and survivors. No demographic or behavioral factor had a statistically significant (Fisher's 2-sided exact $p < 0.05$) or marginally significant ($p < 0.1$) association with mortality. However, there was a trend towards an association with injection drug use (IDU) with 63.6% of decedents vs. 36.2% of survivors having had a history of IDU ($p = .11$) (*Jepsen et al., 2004*).

## DISCUSSION

Homeless and unstably housed youth in San Francisco experience a mortality rate in excess of ten times that of the state's general youth population. The primary causes of death in

**Table 3  Causes of death for cohort decedents.**

| ICD 10 code | Description |
| --- | --- |
| B20.8 | HIV disease resulting in other infectious and parasitic diseases |
| F10.2 | Mental and behavioral disorders due to use of alcohol (Dependence syndrome) |
| F19.1 | Mental and behavioral disorders due to multiple-drug use and use of other psychoactive substances (Harmful use) |
| F19.2 | Mental and behavioral disorders due to multiple-drug use and use of other psychoactive substances (Dependence syndrome) |
| R99 | Other ill-defined and unspecified causes of mortality |
| W80 | Inhalation and ingestion of other objects causing obstruction of respiratory tract |
| X44 | Accidental poisoning by and exposure to other and unspecified drugs, medicaments, and biological substances |
| X64 | Intentional self-poisoning (suicide) by and exposure to other and unspecified drugs, medicaments, and biological substances |
| X74 | Intentional self harm (suicide) by other and unspecified firearm damage |
| X81 | Intentional self harm (suicide) by jumping or lying before moving object |
| X95 | Assault (homicide) by other and unspecified firearm discharge |

our cohort were suicide and/or alcohol- or drug-related, similar to those observed in some previous studies (*Hwang, 2000*; *Roy et al., 2004*; *Evans et al., 2012*).

Although the standardized mortality ratio we found was higher than relative mortality measures from previous US studies that relied on service based sampling (*Barrow et al., 1999*; *Hibbs et al., 1994*; *Hwang et al., 1997*), these SMRs are not directly comparable. As these studies were not all standardized to the same population, differences between effect measures may occur due to differences in the demographic profiles of the study samples, in addition to differences in the stratum specific relative risks of mortality.

The females in our study cohort experienced a non-significant trend towards a higher mean excess risk of mortality than the males. However, all previous studies calculating gender specific SMRs but one (where there were no female decedents) have reported a higher point estimate for the SMR for females than for males (*Nordentoft & Wandall-Holm, 2003*; *Babidge, Buhrich & Butler, 2001*; *Hwang et al., 1997*; *Roy et al., 2004*; *Evans et al., 2012*).

Our study adds to the literature in several ways. To our knowledge this is the first prospective cohort study of the mortality of street youth in the United States, and furthermore, the only such mortality study to have recruited participants entirely through street-based venues, rather than service or shelter-based sampling. Our findings are thus more likely to reflect the risk of mortality of a youth found in a street setting versus a youth who has already accessed services. A further strength of our study is that we adjusted our

SMRs for age, race and sex, unlike most prior studies which have been simply age and sex-adjusted. Given the recognized disparities in US mortality by race/ethnicity, this is an important addition to current analyses (*Mulye et al., 2009*).

The limitations of our study should be considered. Though venue-based sampling allows for a more representative sample of participants than samples recruited through service providers, it remains a non-random sample with limited generalizability. The English language requirement for interview participation may have excluded potential Spanish-speaking or other monolingual participants, impacting the representativeness of this study. However, the venues where we recruited youth were generally not venues frequented by street-based monolingual Spanish-speaking youth, who tend to congregate in neighborhoods distinct from the neighborhoods where the Y-SE study was conducted.

The exclusion of youth under the influence of substances or exhibiting emotional distress or agitation at the point of recruitment may have excluded youth with a higher propensity toward substance abuse or serious mental health issues, thus potentially underestimating deaths. However, we recorded only one youth who was excluded for this reason (due to persistent psychosis).

Our study was conducted in San Francisco, a city with a relative wealth of resources for the homeless (*Department of Housing and Urban Development, 2015*). Thus, although half of the reported deaths occurred outside of California, the experiences and rate of mortality experienced by our participants may not be generalizable to homeless youth from elsewhere in the United States.

Furthermore, we may have missed cases of deaths through incomplete or incorrect identifying information for participants, deaths outside of the United States, or deaths that were not captured in the NDI. However as undercounting deaths in the cohort would lead us to underestimate mortality rates for homeless youth, any bias would be towards the null. Finally, because we were only able to adjust for age, race, and gender, there is possibility of confounding in our SMR estimate.

Though it is not a limitation, a cautionary note is indicated here. The SMRs that we report should not be directly compared to those in the table, nor should the SMRs in the table be compared to each other without a clear understanding that the referring group is different for each of the papers. Thus, we can only state that youth in each location had a higher or lower mortality rate, relative to its own referent population, not to each other's rate.

Despite these limitations, our findings hold important potential policy implications, particularly given the federal commitment to ending youth homelessness by 2020 (*United States Interagency Council on Homelessness, 2016*). While our results, and the results of previous studies, do not conclusively prove that youth homelessness causes excess mortality, it does seem reasonable to assume that homelessness among youth is an indicator of high mortality risk. Even if the association between homelessness and mortality is not causal, the high mortality rate in this cohort of young homeless men and women is a stark reminder that interventions to reduce premature death in street youth are needed. Studies of adults have suggested that providing long-term housing decreases mortality (*Metraux et al., 2011*). Provision of housing is likely to be life-saving for youth as well.

Although rates of IDU were higher among decedents (63.6%) than among those who survived (36.2%), IDU was not significantly associated with mortality ($p = .11$). However, we did not have enough outcomes to definitively determine or refute this relationship. Additional data would be necessary to definitively determine a causal relationship between IDU and the increased risk of death among street-recruited homeless youth. Nevertheless, the high rate of IDU among decedents is consistent with prior findings of a high rate of mortality among injection-drug using youth in San Francisco (*Evans et al., 2012*). Our findings support the development of mental health, harm reduction, and other substance use interventions tailored to street youth as a public health priority for this population (*Seal et al., 2005*). Roy and colleagues in Montreal have suggested that the drop in mortality rates among their cohorts may have been due in part to new services for homeless youth implemented during this time, particularly suicide-prevention services (*Roy et al., 2010*).

Further research is warranted to explore the predictors of mortality among youth, in particular the reasons why females may have a higher SMR than males. Should a gender difference prove to be significant, this may reflect differences in the effect of the street social environment on the health of males versus females (*Bourgois, Prince & Moss, 2004*; *Valente & Auerswald, 2013*). Such information may help inform the development of appropriate, sub-group specific services for this group of highly marginalized, highly vulnerable youth. Given that these youth come disproportionately from groups for which society has a fiduciary responsibility, including survivors of physical and sexual abuse, foster youth and youth with a history of involvement in the juvenile justice system, our collective mandate to address their disparity in mortality is even more pressing.

### Funding

The research described in this paper was conducted with support from the National Institute of Child Health and Development (K-23, HD 0149003), the University of California at San Francisco Research Evaluation and Allocation Committee and Committee on Research (PI: C. Auerswald), Health Resources and Services Administration Title IV/Ryan White Funds (Larkin Street Youth Services), and the University of California, San Francisco Department of Pediatrics Chairman's Funds. The funders had no role in study design, data collection and analysis, decision to publish, or preparation of the manuscript.

### Grant Disclosures

The following grant information was disclosed by the authors:
National Institute of Child Health and Development: K-23, HD 0149003.
University of California at San Francisco Research Evaluation.
Allocation Committee and Committee on Research.
Health Resources and Services Administration Title IV/Ryan White Funds.
University of California, San Francisco Department of Pediatrics Chairman's Funds.

### Competing Interests

The authors declare there are no competing interests.

## Author Contributions

- Colette L. Auerswald conceived and designed the experiments, analyzed the data, wrote the paper, reviewed drafts of the paper.
- Jessica S. Lin wrote the paper, prepared figures and/or tables, reviewed drafts of the paper, prepared and obtained the data from the National Death Index.
- Andrea Parriott analyzed the data, wrote the paper, reviewed drafts of the paper.

## Human Ethics

The following information was supplied relating to ethical approvals (i.e., approving body and any reference numbers):

The protocol was approved by the Committee for Human Subjects of the University of California at San Francisco (Study Number 11-05254).

## Data Availability

The raw data cannot be disclosed due to the fact that the few deaths in the cohort mean that even de-identified individual data could lead to disclosure.

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
