# Peer review of "Six-year mortality in a street-recruited cohort of homeless youth in San Francisco, California"

_PeerJ, doi:10.7717/peerj.1909_

## Round 0.1 · original submission · Major Revisions

Dear Authors, the manuscript has been reviewed by two reviewers who have raised major concerns which need to be addressed prior to any further consideration for publication.

·

Basic reporting

The paper is well written and clearly describes the main findings. The structure of the paper also follows the journal’s guideline. However, some background information is missing, which substantially weakens authors’ justifications for this study. Below are more specific requests for additional information.

1. Line # 58-59: Please elaborate the difference in the context of/resources for youth homelessness between the United States and Canada.

2. Line # 71-73: Has venue-based sampling been validated as a method to generate a more representative sample compared with street-based sampling? If so, please provide that information to justify your selection of the sampling method.

3. Line # 73-74: What would be your expected differences between two samples?

Experimental design

In general, the study design is sound and the authors’ choice of analytic approaches is appropriate. Yet, when they described statistical analyses, some details were missing, which is likely to limit reproducibility of this study. Specific comments are listed as follows:

1. Line # 112: Provide justification of using SMR over direct standardization method.

2. Line # 120-121: Did you use Californian youth as a reference population? If so, please specify an age restriction applied to the reference population.

3. Line # 120-121: I think that youth in San Francisco should be used as a reference group, which is a better counterfactual for the study population than Californian youth. At the minimum, the authors should explain why you decide to choose Californian youth over San Francisco youth for your reference group.

4. Line # 145-147: Please describe a type of statistical test you have performed to draw p-value. Is it based on comparing 95% CI of two IDU-specific SMRs?

Validity of the findings

Because SMR is specific to the reference population, it is not recommended to compare SMRs across different studies. For example, supposed that you estimate age- and sex-adjusted SMR among homeless youth in US. You cannot compare this with SMR among homeless youth in Canada, unless age- and sex-specific mortality rates are parallel between Canadian and US youths (i.e., age- and sex-specific mortality rates are independent of locations of two reference populations). For further discussion about this limitation, please read “Breslow NE, Lubin JH, Marek P, et al. Multiplicative models and cohort analysis. J Amer Stat Assoc 1983;78:1-12.”

1. I suggest the authors acknowledging this limitation when they compare SMRs in the current study with those in the previous studies.

2. Line # 157-159: Please explain why higher SMRs for males vs. females are uniquely observed in this study. What would be plausible reasons for the inconsistent finding between this and previous studies?

3. Table 1: Please include the information about reference populations.

Reviewer 2 ·

Basic reporting

Could it be specify the age mean range of youth in the eleven studies, which have presented data for a youth subgroup (line 47)?

The authors wrote the context of and resources for youth homelessness in the US and Canada were different (line 60-61). Could it be briefly specify in which ways these two countries differed in context of and resources for youth homelessness?

The authors cited eleven studies which have presented data for a youth subgroup in first part. Later in text, the author summarizes the exiting published findings regarding street youth but it is not the same eleven references. The table does not include studies 17,19 as described in the references line 49. But it include references 8, 25 and 26, why?

In the sentence line 65-66 “Table 1 summarizes the exiting published findings regarding street youth mortality in Europe and North America”, the author could add “in Australia” for the study in Sydney.

The table 1 could be more used in description in first part of the introduction. I suggest to refer to this table in line 48.

The authors did not respect the suggested format of the journal for references (it is "Name.Year" and not superscript) and for names of the standard sections (background instead of introduction etc…).

Experimental design

The authors have previously described recruitment, sampling but it seems important to know briefly the objectives of the Street Y-SE study. Does this study was conducted to examine mortality in this population or it has general descriptive purposes on health events? In this case, a brief sentence could be adding on others type of collected variables in the Street Y-SE study (if there are others variables).

Homeless who were under influence of substance or emotionally distressed were not included. In a previous study of the authors (influence and predictors of onset of injection drug use in this population), I understand the need to exclude homeless who were under influence of illicit substances. However, in this present study, it could be an important bias. This implies a strong underestimation of mortality since people with addictive behavior or with mental disorders have a higher mortality than others people. Secondly, why the authors have excluded homeless who were too distressed or agitate? These behaviors could reveal mental disorders and could underestimate the mortality once again.

What is the language of the interview? It could be specify in methods and develop in discussion regarding the representativeness of the sample.

In the abstract, authors said there were 266 participants, but in results, they indicated 218 youth. Could they explain the difference?

Regarding the causes of death of homeless, the authors have considered all causes of death or the leading causes of death?

Validity of the findings

In discussion, the authors should dealing with underestimating of mortality due to exclusion criteria (being under the influence of substances or emotionally distressed).

The authors said there were probably missed cases of deaths through incomplete or incorrect identifying information for participant. They are certainly right. However, they said after this “we propose that any bias would be towards the null”. Could they develop this part? What they think about the prevalence of unidentified bodies among homeless deaths in San Francisco?

Additional comments

The authors performed original works among a hard-to-reach population and a high-risk population regarding mortality and health events. The article is well structured. The survey method is interesting and has the merit of being published. Even if they were limited by the small size of sample, the results adds to the literature on mortality in this population who is few investigated. Some elements in methods and discussion should be developed. This kind of study is necessary in the implementation of public policies to improve the status of this high-risk population.

---

## Round 0.2 · Major Revisions

Dear authors,

I have sent your revised manuscript to the previous reviewers and one of them has indicated your paper needs more corrections. Please, try to solve the highlighted issues to consider your paper for publication.

With respect and warm regards,
Dr Palazón-Bru (academic editor for PeerJ)

·

Basic reporting

The authors have addressed comments on the original manuscript. There is no further comment on the aspect of basic reporting.

Experimental design

There is no further comment.

Validity of the findings

The authors have properly addressed earlier concerns on comparisons of SMRs across different studies. There is no further comment on validity of the findings.

Reviewer 2 ·

Basic reporting

1. For your information, a recent study presented data for a youth subgroup. 28,033 adults seen at Boston Health Care for Homeless Program, whose 3,491 homeless were 18-24 y/o. They used age-standardized incidence ratios (SIRs) and age-standardized mortality ratios but they did not calculate SMR per age strata. Reference : Baggett TP, Chang Y, Porneala BC, Bharel M, Singer DE, Rigotti NA. Disparities in Cancer Incidence, Stage, and Mortality at Boston Health Care for the Homeless Program. Am J Prev Med. 2015 Nov;49(5):694–702.

2. I think the sentence line 48 : “Table 1 summarizes […] and Australia” could be moved at the line 46, before the sentence “Of the studies that have examined…”

3. At line 69, the sentence could be reformulated such as “A limitation of almost of the studies among homeless, such as those listed in Table 1, is the use of…”

Experimental design

4. Line 84: The objectives of the Street Y-SE study were not clearly described. Would it be exact if the sentence: “The Street Youth in Social Environments (Street Y-SE) study was a longitudinal study of San Francisco street youth, their social networks, street culture and HIV risk”, was replaced by “The Street Youth in Social Environments (Street Y-SE) study was a longitudinal study which examined the relationships among street culture, social networks and STI/HIV risk in homeless youth in San Francisco” ?

5. Lines 89-95: this paragraph “Prior to enrollment […] redundancy” is not useful for understanding the context of this present study if it was previously described.

6. In the previous review, I wrote “Regarding the causes of death of homeless, the authors have considered all causes of death or the leading causes of deaths”. I apologize for the misunderstanding. I meant, in some countries, the medical officer has to indicate, on certificate death, the underlying cause of death that initiated the morbid process leading to death. In addition, there may have been causes associated with the death but were not directly related to the disease causing it. Is it the case in this study or not?

7. Lines 188-192: The authors had accurate the language for interview participation. They said there were few Spanish-speaking youth or other monolingual participants: “we recruited youth were generally not venues frequented by street-based monolingual Spanish-speaking youth”. Do they have more information about it? Could they quantify the proportion of no English-speaking among homeless youth?

Validity of the findings

8. Lines 147-148: Since the number of deaths was 11, it seems not relevant to put proportions. The authors could write only the number (as it was originally).

9. Lines 159-160: I disagree with the authors when they used the term “marginal” to describing the association with IDU and mortality. If the p-value is inferior to 0.05, then there is no association. This is the same remark for the paragraph in discussion about this association (lines 222-223).
In the same way, lines 172-173, the authors said that women had a higher mean excess risk than men whereas the gender-specific SMRs were not significantly different. It seems better to nuance this sentence with the idea of the trend such as: “Though females in our study cohort tend to experience a higher mean excess risk…” This is the same remark for the paragraph in discussion about this association (lines 233-234), the authors cannot say “the reasons why female have a higher SMR than males”, because in this study, there was a trend but it was not significant. Thus at this stage, there was no difference between the gender-specific SMRs.

10. Line 198: The authors said that San Francisco is a “city with a relative wealth of resources for the homeless”. Do they have some references which indicate that SF was more wealthy of resources for the homeless than other cities in California (or in the USA)?

11. Line 205: I do not understand the meaning of “any bias would be towards the null”. Do the authors mean that since there was possibly an undercounting deaths, then the risk of overestimate was null?

Additional comments

The authors performed original works among a hard-to-reach population and a high-risk population regarding mortality and health events. The article is well structured. The survey method is interesting and has the merit of being published. Even if they were limited by the small size of sample, the results adds to the literature on mortality in this population who is few investigated. Some elements in methods and discussion should be developed. This kind of study is necessary in the implementation of public policies to improve the status of this high-risk population.
In the first review, the authors have correctly and respectfully answered to the revisions. I still make some remarks in this review, mainly on elements in method and discussion.

---

## Round 0.3 · accepted · Accept

Dear authors,

After analyzing the answers for the previous questions, your manuscript has high standards to be published in PeerJ.

Congratulations!

With respect and warm regards,
Dr Palazón-Bru (academic editor for PeerJ)